# Factors associated with dietary diversity among pregnant women in the western hill region of Nepal: A community based cross-sectional study

**Vintuna Shrestha**[1]*, **Rajan Paudel**[2], **Dev Ram Sunuwar**[3], **Andrew L. Thorne Lyman**[4], **Swetha Manohar**[4,5], **Archana Amatya**[2]

1 Department of Nursing, Dhaulagiri Prabhidhik Shikshya Pratisthan, Council for Technical Education and Vocational Training, Baglung, Nepal, 2 Central Department of Public Health, Institute of Medicine, Tribhuvan University, Kathmandu, Nepal, 3 Department of Nutrition and Dietetics, Armed Police Force Hospital, Kathmandu, Nepal, 4 Center for Human Nutrition, Department of International Health, Johns Hopkins Bloomberg School of Public Health, Baltimore, MD, United States of America, 5 International Development Program, Nitze School of Advanced of International Studies (SAIS), Johns Hopkins University, Washington DC, United States of America

* shresthavintuna@gmail.com

## Abstract

### Background

Dietary diversity can play an important role in providing essential nutrients for both mother and fetus during pregnancy. This study aimed to assess the factors associated with dietary diversity during pregnancy in the western hill region of Nepal.

### Methods

A cross-sectional study of 327 pregnant women was conducted in an urban municipality of Baglung district in the western hill region of Nepal. A semi-structured questionnaire was used to collect information on household demographic and socioeconomic status, food taboos, household food security status, nutrition-related knowledge in pregnancy, and women's empowerment. Women consuming ≥5 of 10 food groups in the past 24 hours were defined as consuming a diverse diet using the Minimum Dietary Diversity Score for Women (MDD-W) tool. Bivariate and multivariate logistic regression was used to estimate crude odds ratio (cOR) and adjusted odds ratios (aOR) and 95% confidence intervals (CIs) to understand factors associated with dietary diversity.

### Results

Almost 45% (95% CI: 39.6–50.4) of the participants did not consume a diverse diet and the mean dietary diversity score was 4.76 ± 1.23. Multivariable analysis revealed that women with greater empowerment (aOR = 4.3, 95% CI: 1.9–9.9), from wealthier households (aOR = 5.1, 95% CI: 2.7–9.3), joint families (aOR = 2.7, 95% CI: 1.4–5.1), employment (aOR =

**Data Availability Statement:** All relevant data are within the manuscript and its Supporting Information files.

**Funding:** Support for this effort was provided by the Feed the Future Innovation Lab for Nutrition which is funded by the United States Agency for International Development (USAID) award AID-OAA-L-10-00006. Additional supportive funds came from the Gates Foundation and the Sight & Life Foundation. The funders did not play any role in the study design, data collection, and analysis, decision to publish, or preparation of the manuscript.

**Competing interests:** The authors have declared that no competing interest exist.

2.2, 95% CI: 1.2–4.1), and had adequate nutrition knowledge (aOR: 1.9, 95% CI 1.1–3.4) had higher odds of dietary diversity.

## Conclusion

Along with socioeconomic status, women's empowerment and nutrition knowledge were modifiable risk factors that should be considered as targets for programs to improve women's health during pregnancy.

## Introduction

Pregnancy is a life stage involving considerable physiologic and metabolic changes [1]. Diet becomes even more important during pregnancy given the increased micronutrient, energy, and macronutrient requirements to support the health of the mother, the healthy fetal growth [2, 3], and the accumulation of stores for lactation [4]. The right nutrition during pregnancy supports adequate intrauterine growth of the fetus and normal birth weight, which can have life-long consequences for development [5, 6].

Micronutrient deficiencies in pregnancy are particularly important given their high prevalence and public health consequences [2, 7, 8]. Dietary diversification is an important strategy to address micronutrient and macronutrient deficiencies [9]. Dietary diversity indicators, measured as the number of different foods or food groups consumed over a specific reference period [10] are increasingly used as an easy-to-measure proxy for dietary quality and nutrient adequacy in pregnancy [11, 12] and lactation [13].

Pregnant women in low and middle-income countries are particularly at risk of micronutrient deficiencies as their diets are monotonous, predominantly cereal-based and often include minimal consumption of nutrient-dense animal source food, vegetables, and fruits [14]. It has been estimated that about two-thirds of pregnant women suffer from nutritional anemia in developing countries [7]. Especially, South Asian countries accounts for the largest burden of anemia with estimated prevalence of 52.5% among women of reproductive age (WRA) [15]. Also, there is growing evidence of increasing dietary calcium deficiency during pregnancy [8]. The burden of micronutrient deficiencies such as zinc, iodine, and vitamin B12 is particularly high in South Asia and sub-Saharan Africa. It is noted that pregnant women may be at greater risk for inadequate intake than non-pregnant and non-lactating women [16]. There is increasing evidence of nutritional deficiencies among women of reproductive age in Nepal [17, 18].

According to the 2016 Nepal Demographic Health Survey (NDHS), the Nepalese diet is monotonous and mainly cereal-based, with low consumption of food from animal sources, vegetables, and fruits [19]. Similar dietary patterns among women of reproductive age were reported by Nepal National Micronutrient Status Survey (NMSS) 2016 [17]. Additional evidence has shown dietary diversity scores to be low (mean DDS = 4.1±1.2) among mothers [20]. None of the studies mentioned above have focused on evaluating dietary diversity among pregnant women which indicates a lack of contextual information about dietary diversity in this group of the population.

Many studies outside of Nepal have identified common factors for low dietary diversity including low education, low family income [21, 22], household food shortage [23], food taboos [24], lack of nutritional knowledge [25], and low women empowerment level [26].

Despite understanding the importance of higher dietary quality, especially during pregnancy and other stages of life, maternal malnutrition continues to exist in developing countries

at high rates [14]. To the best of our knowledge, dietary diversity and its associated factors among pregnant women have not been assessed comprehensively in the Nepalese context. Although some studies have been done in the plains (Terai) region of the country [27, 28], the quality of the diets of pregnant women in the hills region of the country, an area with very different sociocultural and economic realities, remains understudied. To fill this knowledge gap, we conducted a study to determine the prevalence of low dietary diversity and associated factors among Nepalese pregnant women residing in the urban municipality of western hill region.

## Materials and methods

### Study design and setting

A community-based cross-sectional study was conducted from September to November, 2018, among pregnant women residing in the hill region of Baglung district in Gandaki province, Nepal. According to the 2011 census, Baglung municipality is one of the largest municipalities comprising of 14 wards (sub-districts) [29] with a total population of 57,823 individuals [30] located in western Nepal, and also administrative headquarter of the Baglung district and Dhaulagiri Zone [31] (Fig 1). The percentage of low birth weight in Gandaki province (10.4) is similar to national average (11.7) [32]. In Baglung district only 47% of pregnant women had 4 ANC visit as per the protocol which is lower than the national average of 69%. Also, percentage of pregnant women receiving iron folic acid supplementation is lower (46.8%) compared to national 91% [19, 29].

### Sample size and sampling strategy

The sample size was calculated using the single population proportion for finite population formula [33]. An expected pregnancy of 1,818 was used as the finite population (N) [29]. The proportion (p) of women consuming diverse diet was estimated as 58% [28] based on a study conducted in Nepal with an allowable error (d) of 5% and standard normal deviation (Z) at confidence limit of 95%. The final calculated sample size was 327 with addition of 5% non-response rate.

Baglung Municipality was purposively selected. Out of 14 wards, five were randomly selected. The selected wards included Narayansthan, Amalachaur, Bhakunde, Tityang, and Kudule. A complete list of pregnant women at the ward level was created with the help of an Antenatal Checkups (ANC) register maintained at the health post and maintained by the Female Community Health Volunteers (FCHV) of the respective wards. In every local administrative unit, there is a provision of at least one FCHV, who works voluntarily to promote maternal newborn child health and family panning in the community [34]. The total pregnant woman identified at the start of the study was 678. Pregnant women were listed and a random number table was created for 327 samples. As per the random number generated pregnant women were selected and enrolled in the study.

### Data collection and variables

Data were collected through individual interviews at the respondents' homes conducted in Nepali by trained Master in Public Health Nutrition students trained for five days on the objective of the study, data collection procedure, sampling method, ethical aspects of the study, data entry techniques, and data management.

**Outcome variable.**   The dietary diversity score of pregnant women was measured using a 24 hour dietary recall following the guidelines outlined for calculation of the Minimum

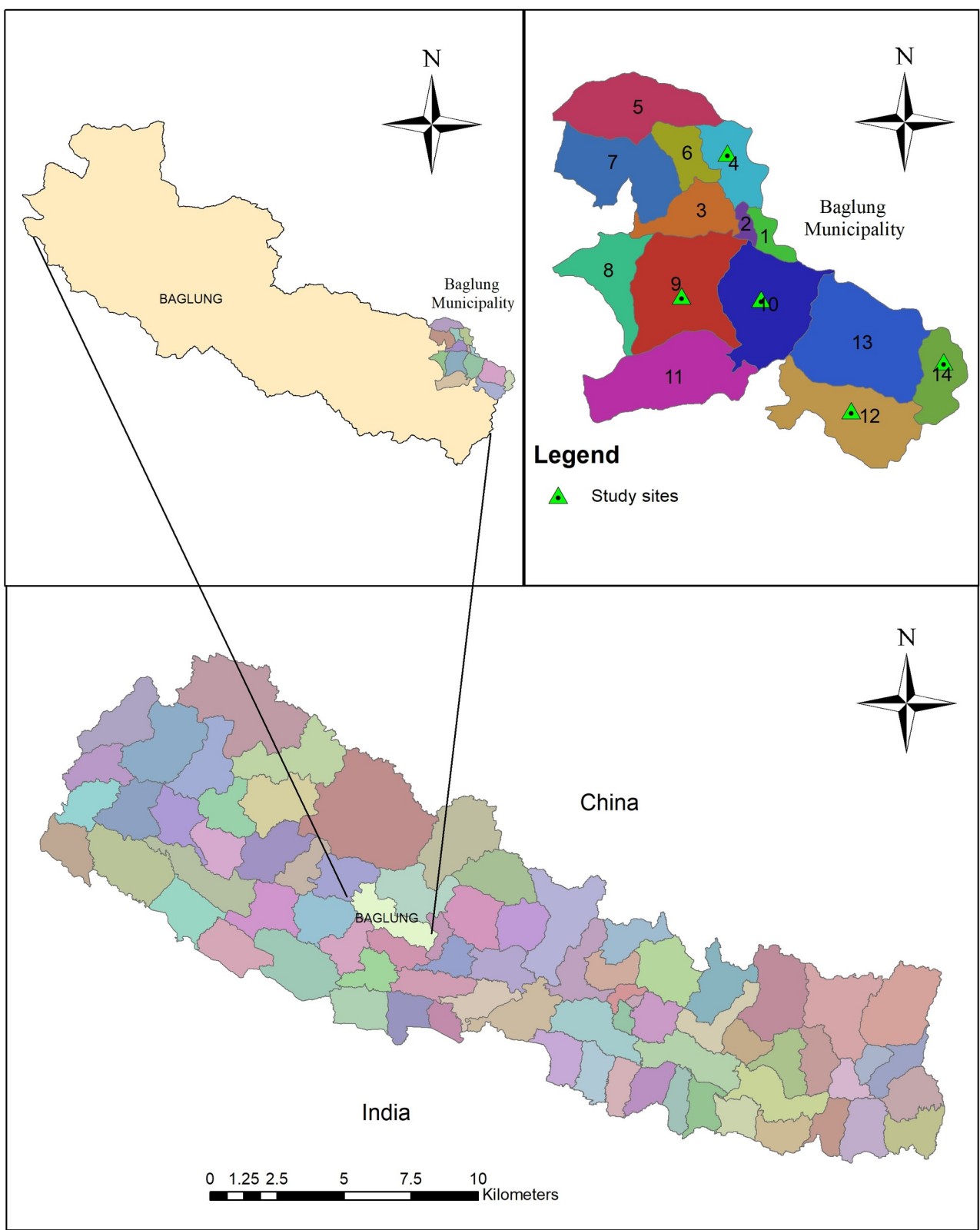

**Fig 1. Map showing the location of study sites.**

Dietary Diversity for Women of Reproductive age (MDD-W) indicator [35]. During the 24 hour recall, the participants were asked to describe food and drinks consumed in the past 24 hours and each food was recorded only when the consumed amount was greater than 15g which was ensured using a food atlas [36]. Ten food groups, listed in S1 Table, were included in the Dietary Diversity Score (DDS) estimated by summing food groups consumed. Food groups that were consumed received a score of '1'. The cumulative DDS was categorized to construct a dichotomized outcome variable: dietary diversity (consuming five or more food groups) and no dietary diversity (consuming less than five of the ten food groups).

A five step multi-pass method was used in conjunction with a photographic food atlas. This approach has been shown to reduce underreporting [37]. In brief, this method included the following five different passes to describe food intake over the previous 24 hours: (1) collect a free recall, using non-specific probes, starting from when the respondent woke up the previous morning (2) probe using a standard list of commonly forgotten foods (such as tea, drinks, and fruit) (3) ask for the time and place that each item was consumed (4) collect portion size information using the atlas and clarify the exact food types (5) use a series of final probes (referring to snacks and food eaten outside the home) and recap all the recorded foods in chronological order.

**Predictor variables.** Information on socio-demographic variables such as participants' age, ethnicity, religion, family structure, participants and spouse education and employment status were collected. A joint family typically consists of three or more generations and their spouses living together as a single household. In this study joint family refers to the one where woman lives with their in-laws and other relatives. Similarly, information related to beliefs on food taboos was collected using a semi-structured questionnaire. The indigenous caste Janajati, included the disadvantaged groups. The Dalit and minority groups (such as Madhesi and Muslim communities) were also included as disadvantaged groups that were considered lower in the caste based stratification within Nepalese society while, the Brahmin/Chhetri were included as advantaged ethnic groups [38].

A wealth score was developed based on a Principal Component Analysis (PCA) of assets owned and economic status using methods recommended in the Nepal Demographic and Health Survey (DHS) [19]. More than twenty variables including household related variables such as main fuel for cooking, availability of separate kitchen, possession of land, main material of wall, roof and floor; some household assets were included to measure the wealth index. The wealth score was treated in tertiles for the purposes of the analysis.

Questions on nutritional knowledge were adapted from the FAO Knowledge Attitude and Practice Manual [39] and a cross-sectional study on pregnancy-specific knowledge [40]. Variables included in S2 Table were included in assessing knowledge on nutrition during pregnancy. Demonstrated knowledge for each knowledge variable was given a score 1. The total score ranged from 0 to 9. A pregnant woman was classified as having adequate knowledge if her knowledge score was 6 and above and inadequate knowledge if her score was <6.

The questionnaire for women's empowerment was adapted from Nepal's DHS [41] and a country-relevant cross-sectional study on women's empowerment [42]. Women's empowerment was assessed using a women's empowerment index which reflects: women's involvement in household decision making, membership in community group, cash earning, ownership of household/land, and educational status. Variables included in construction of women empowerment index are shown in S3 Table. Women's empowerment was classified as low, moderate and high, but during analysis the low and moderate groups were merged and creating a dichotomous variable. The Household Food Insecurity Access Scale (HFIAS) developed by the Food and Nutrition Technical Assistance (FANTA) project

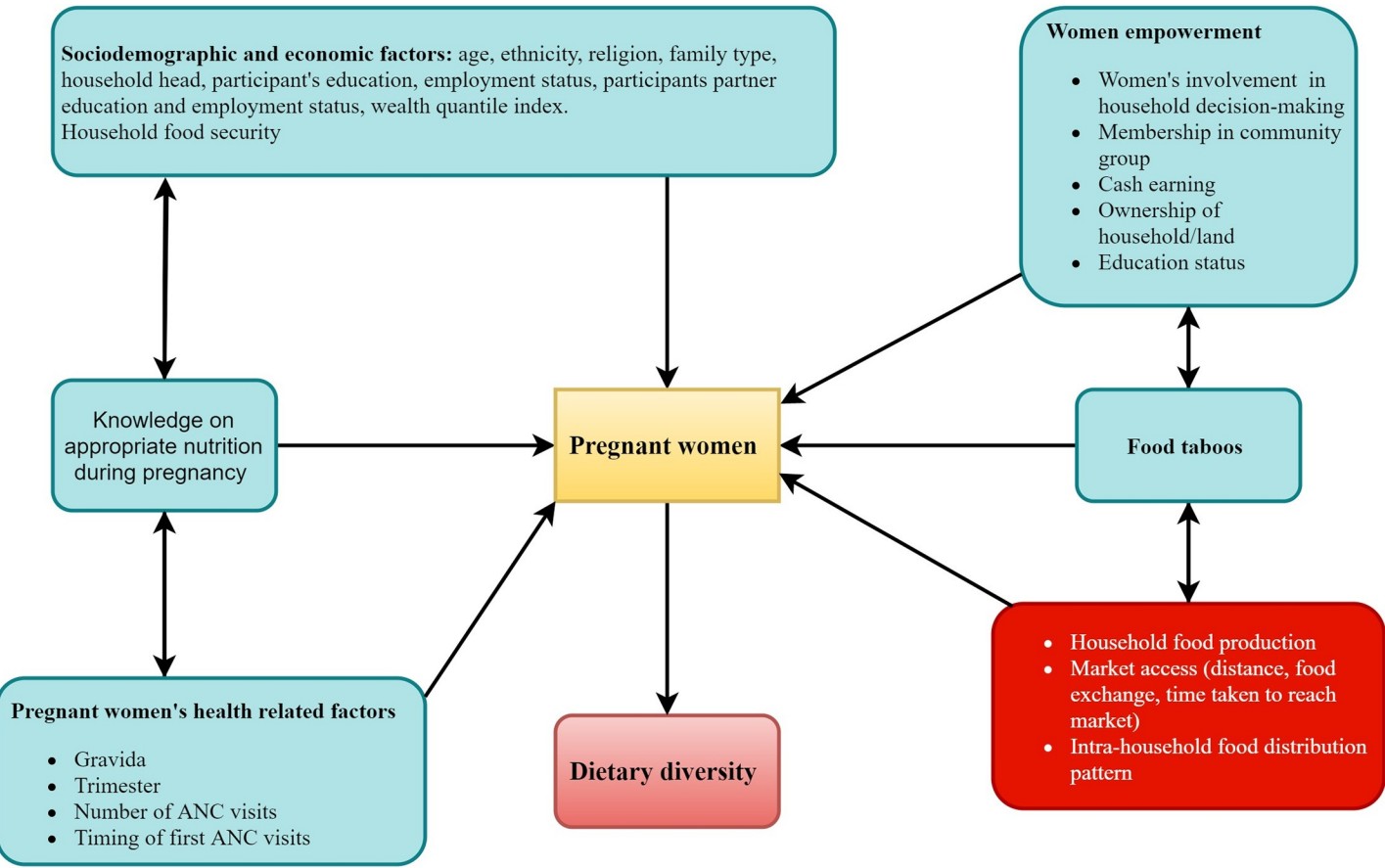

**Fig 2. Conceptual framework on factors associated with dietary diversity.**

was used to assess experience of household food insecurity, specifically the access to food domain of food insecurity [43] (Fig 2).

## Data management and analysis

Data checking, compiling and editing were done manually to ensure completeness and accuracy before data were entered for analysis. Data were entered in Epi Data version 3.1 and analyzed using Stata/MP version 14.1 (Stata Corp LP, College Station, Texas).

Exploratory data analysis included frequency estimations and cross-tabulations of each predictor and outcome variable. Categorical variables were presented in percentage and frequency whereas continuous variables were presented in mean and standard deviation. In bivariate analysis, Chi-square test (or Fisher exact test) was applied to test the significance of the association between predictors and outcome variables. Binary logistic regression was used to estimate odds ratios and 95% Confidence Intervals (CIs). Variables that were found statistically significant (95% C.I. not overlapping the null) during bivariate analyses were checked for multicollinearity by calculating the Variance Inflation Factor (VIF) [44] and backward stepwise logistic regression analysis was run adjusting for variables significant $P<0.05$ in the unadjusted analysis. Crude odds ratio (cOR) and adjusted odds ratio (aOR) with 95% CI were calculated to assess the presence and strength of

association, with a threshold of p<0.05 used for determination of statistical significance. The VIF report is presented in S4 Table.

## Ethical considerations

The ethical approval for this study was obtained from the Institutional Review Committee at the Institute of Medicine Ref: 73(6.4.E)2/075/076. The local municipality office and health posts of selected wards were also asked for consent to enter and conduct interviews. Informed written consent was obtained from each of the participants before proceeding to data collection. Participants were informed about voluntary participation, their right to refuse participation at any point, and the confidentiality of their identity.

# Results

## Demographic characteristics

Table 1 depicts the descriptive characteristics of the participants. The mean age of study participants was 23.6 years. The majority (74%) of participants belonged to a joint family and about

**Table 1. Demographic characteristics of the participants (n = 327).**

| Variables | Number (n) | Percent (%) |
|---|---|---|
| **Socio-demographic characteristics** | | |
| **Age (years), Mean ± Standard Deviation (SD)** | 23.6 ± 0.2 | |
| ≤24 years | 214 | 65 |
| >24 years | 113 | 35 |
| **Ethnicity** | | |
| Dalit and minorities | 63 | 19 |
| Janajati | 87 | 27 |
| Brahmin/chhetri | 177 | 54 |
| **Religion** | | |
| Hindu | 314 | 96 |
| Non hindu | 13 | 4 |
| **Joint/extended** | 241 | 74 |
| **Household head** | | |
| Female | 102 | 31 |
| Male | 225 | 69 |
| **Participant education level** | | |
| No education | 11 | 3.4 |
| Primary level | 24 | 7.3 |
| Some secondary level | 29 | 8.9 |
| SLC or above | 263 | 80.4 |
| **Participant partner education** | | |
| Secondary or lower | 69 | 21 |
| SLC or above | 258 | 79 |
| **Participants employment** | | |
| Employed | 92 | 28.1 |
| Unemployed | 235 | 71.9 |
| **Participants partner employment** | | |
| Employed | 294 | 90 |
| Unemployed | 33 | 10 |

(*Continued*)

**Table 1.** (Continued)

| Variables | Number (n) | Percent (%) |
|---|---|---|
| **Land ownership** | | |
| No | 188 | 57 |
| Yes | 139 | 43 |
| **Women empowerment** | | |
| Low/ Moderate | 246 | 75 |
| High | 81 | 25 |
| **Primipara** | 185 | 57 |
| **Pregnancy trimester** | | |
| First | 21 | 7 |
| Second | 135 | 41 |
| Third | 171 | 52 |
| **Timing of first ANC visit at <4 months of gestation** | 306 | 94 |
| **ANC visits (<4)** | 249 | 77 |
| **Food and nutrition factors** | | |
| **Food taboo practice** | | |
| Yes | 98 | 30 |
| None | 229 | 70 |
| **Nutritional Knowledge** | | |
| Inadequate | 181 | 55 |
| Adequate | 146 | 45 |
| **Food security** | | |
| Food insecure | 26 | 8 |
| Food secure | 301 | 92 |

two-third of the participants were from male-headed households. Regarding education, majority of the participants had completed SLC or higher levels of education. Approximately, three-fourths of women were classified as low/moderately empowered. More than half (57%) of the participants were primiparous. Less than half (45%) of participants had adequate knowledge regarding appropriate nutrition during pregnancy. Only 8% of the participants belonged to food-insecure households.

A third of women reported adhering to certain food taboos during pregnancy. Commonly avoided foods by women included (33.7%), jackfruit, (23.5%) papaya, (15.3%) honey and (31%) other prohibited foods were chilies, horse gram, squash, fish, banana, coconut, egg yolk, *anadi peetho* (a type of rice flour), peanuts, bamboo shoots, Sichuan pepper, mushroom, turmeric, sugarcane, jaggery (unrefined sugar made of concentrated sugarcane juice), and *phurse* (a type of pumpkin).

## Dietary diversity status

Consumption of a diverse diet was split near evenly between study participants with44.9% (95% CI: 39.6–50.4) of women consuming a not diverse diet (Table 2). The mean dietary diversity score was 4.76 with SD ± 1.23. Consumption of grains, white roots, tubers and, pulses was reported by all women, whereas very few participants (3.1%) consumed foods from the nuts and seeds group.

Among animal source foods, the majority of women consumed milk and milk products (61.2%) followed by meat, poultry and fish (25.7%), and, eggs (14.1%). Less than half of the

**Table 2. Dietary diversity and individual food group consumption by participants.**

| Characteristics | Number (n) | Percent (%) |
|---|---|---|
| **Food groups** | | |
| Grains, white roots and tubers | 327 | 100.0 |
| Pulses | 322 | 98.5 |
| Nuts and Seeds | 10 | 3.1 |
| Milk and milk products | 200 | 61.2 |
| Eggs | 46 | 14.1 |
| Meat, Poultry and fish | 84 | 25.7 |
| Dark green Leafy Vegetables | 151 | 46.2 |
| Vitamin A rich fruit and Vegetables | 25 | 7.6 |
| Other Vegetables | 230 | 70.3 |
| Other fruits | 163 | 49.8 |
| **Dietary diversity status** | | |
| Diverse % (95% CI) | 180 | 55.0 (49.6–60.4) |
| Not diverse % (95% CI) | 147 | 44.9 (39.6–50.4) |
| Mean ± SD | 4.76 ± 1.23 | |

participants consumed dark green leafy vegetables and other fruits whereas about two-third of the participants consumed other vegetables. However, only 7.6% of women consumed vitamin A rich fruits and vegetables.

## Factors associated with dietary diversity during pregnancy

In bivariate analyses, family type, education level of participant, education level of participants' partner, participant employment status, wealth status, land ownership, number of ANC visit, food taboos, empowerment level, and nutrition knowledge scorewere found to be significantly positively associated with increased odds of dietary diversity.

In multivariable logistic regression analysis, the adjusted odds of consuming a diverse diet was 2.7 times higher among the pregnant women who were from joint/extended family compared to those from nuclear family (aOR = 2.7, 95% CI: 1.4–5.1) (Table 3). Similarly, it also showed the odds of consuming diverse diet were twice as high among the employed pregnant women than unemployed ones (aOR = 2.2, 95% CI: 1.2–4.1). Women from wealthier households were 5.1 times more likely to consume diverse diet than the women from low wealth households (aOR = 5.1, 95% CI: 2.7–9.3). Women with adequate knowledge of nutrition during pregnancy were 1.9 times more likely to consume a diverse diet than the women with inadequate knowledge on appropriate nutrition during pregnancy (aOR: 1.9, 95% CI 1.1–3.4). Women who reported experiencing higher levels of empowerment (i.e. who have a say in household decision making in terms of household purchase, seeking health service and visiting relatives, who are educated, have a regular earning, owns household or land singly or jointly and is a member of any community group) were 4 times more likely to consume diverse diet than the women with low empowerment (aOR = 4.3, 95% CI: 1.9–9.9).

## Discussion

The current paper contributes to a limited but growing literature on socioeconomic, food taboos, nutritional knowledge, and women empowerment related factors and their relationship with dietary intake during pregnancy. We found that several factors including family type, cash earnings, economic status, nutritional knowledge, and women's empowerment play

**Table 3. Bivariate and multivariate binary logistic regression of factors associated with dietary diversity among pregnant women (n = 327).**

| Variables | Dietary diversity | | Bivariate analysis | | Multivariable analysis | |
|---|---|---|---|---|---|---|
| | Not Diverse n (%) | Diverse n (%) | cOR (95% CI) | P-value[1] | aOR (95% CI) | P-value[2] |
| **Age groups** | | | | | | |
| ≤24 years | 94 (43.9) | 120 (56.1) | Ref | 0.607 | - | |
| >24 years | 53 (46.9) | 60 (53.1) | 0.9 (0.5–1.4) | | | |
| **Ethnicity** | | | | | | |
| Dalit and minorities | 37 (58.7) | 26 (41.3) | Ref | | | |
| Janajati | 37 (42.5) | 50 (57.5) | 1.9 (0.9–3.7) | 0.051 | 1.3 (0.6–3.1) | 0.450 |
| Brahmin/chhetri | 73 (41.2) | 104 (58.8) | 2.0 (1.1–3.6) | 0.018* | 0.4 (0.5–2.4) | 0.708 |
| **Family type** | | | | | | |
| Nuclear | 49 (56.9) | 37 (43.1) | Ref | | Ref | |
| Joint/extended | 98 (40.7) | 143 (59.3) | 1.9 (1.1–3.1) | 0.010* | 2.7 (1.4–5.1) | 0.003* |
| **Household head** | | | | | | |
| Female | 41 (40.2) | 61 (59.8) | 1.3 (0.8–2.1) | 0.245 | - | |
| Male | 106 (47.1) | 119 (52.9) | Ref | | | |
| **Participant education level** | | | | | | |
| Secondary or lower | 24 (68.6) | 11 (31.4) | Ref | | Ref | |
| SLC or above | 123 (42.1) | 169 (57.9) | 2.9 (1.4–6.3) | 0.004* | 0.9 (0.3–2.6) | 0.923 |
| **Participant's partner education** | | | | | | |
| Secondary or lower | 43 (62.3) | 26 (37.7) | Ref | | Ref | |
| SLC or above | 104 (40.3) | 154 (59.7) | 2.4 (1.4–4.2) | 0.001* | 1.6 (0.7–3.4) | 0.216 |
| **Participant employment** | | | | | | |
| Employed | 29 (31.5) | 63 (68.5) | 2.1 (1.3–3.6) | 0.003* | 2.2 (1.2–4.1) | 0.009* |
| Unemployed | 118 (50.2) | 117 (49.8) | Ref | | Ref | |
| **Participant's partner employment** | | | | | | |
| Employed | 130 (44.2) | 164(55.8) | 1.3 (0.6–2.7) | 0.425 | - | |
| Unemployed | 17 (51.5) | 16(48.5) | Ref | | | |
| **Wealth status** | | | | | | |
| Poor | 75 (57.3) | 56 (42.7) | Ref | | Ref | |
| Middle | 38 (58.4) | 27 (41.6) | 0.9 (0.5–1.7) | 0.872 | 0.9 (0.4–1.9) | 0.935 |
| Rich | 34 (25.9) | 97 (74.1) | 3.8 (2.2–6.4) | <0.001* | 5.1 (2.7–9.3) | <0.001* |
| **Land ownership** | | | | | | |
| No | 95 (50.5) | 93 (49.5) | Ref | | Ref | |
| Yes | 52 (37.4) | 87 (62.6) | 1.7 (1.0–2.6) | 0.019* | 1.1 (0.5–2.0) | 0.840 |
| **Women empowerment** | | | | | | |
| Low/moderate | 132 (53.7) | 114 (46.3) | Ref | | Ref | |
| High | 15 (18.5) | 66 (81.5) | 5.1 (2.7–9.4) | <0.001* | 4.3 (1.9–9.9) | <0.001* |
| **Maternal Health factors** | | | | | | |
| **Gravida** | | | | | - | |
| Primipara | 79 (42.7) | 106 (57.3) | Ref | | | |
| Multipara | 68 (47.9) | 74 (52.1) | 0.8 (0.5–1.2) | 0.35 | | |
| **Pregnancy trimester** | | | | | | |
| First | 9 (42.9) | 12 (57.1) | Ref | | - | |
| Second | 59 (43.7) | 76 (56.3) | 0.9 (0.3–2.4) | 0.942 | | |
| Third | 79 (46.2) | 92 (53.8) | 0.8 (0.3–2.1) | 0.772 | | |
| **ANC visits** | | | | | | |
| <4 times | 119 (47.8) | 130 (52.2) | Ref | | Ref | |
| ≥4 times | 26 (34.2) | 50 (65.8) | 1.7 (1.0–3.0) | 0.038* | 1.6 (0.8–3.1) | 0.125 |

*(Continued)*

**Table 3.** (Continued)

| Variables | Dietary diversity | | Bivariate analysis | | Multivariable analysis | |
|---|---|---|---|---|---|---|
| | Not Diverse n (%) | Diverse n (%) | cOR (95% CI) | P-value[1] | aOR (95% CI) | P-value[2] |
| **Food and nutrition factors** | | | | | | |
| **Food taboos** | | | | | | |
| No | 115 (48.7) | 121 (51.3) | Ref | | Ref | |
| Yes | 32 (35.2) | 59 (64.8) | 1.7 (1.0–2.8) | 0.028* | 1.5 (0.8–2.7) | 0.158 |
| **Nutritional knowledge** | | | | | | |
| Inadequate | 101 (55.8) | 80 (44.2) | Ref | | Ref | |
| Adequate | 46 (31.5) | 100 (68.5) | 2.7 (1.7–4.3) | <0.001* | 1.9 (1.1–3.4) | 0.013* |

*Denotes the statistically significant at p<0.05.

[1]Bivariate binary logistic regression analysis.

[2]Backward Stepwise logistic regression analyses was run adjusting for variables significant.

P<0.05 in the unadjusted analysis; cOR: crude odds ratio; aOR: adjusted odds ratio; ANC: antenatal checkups; CI: confidence interval.

a key role in facilitating the consumption of diverse diet during pregnancy. We found a particularly strong association between women's empowerment and dietary diversity in this setting, which suggests the potential importance of addressing women's empowerment in nutrition-related programs.

Our study revealed that pregnant women in this part of rural Nepal consume a diet low in diversity. There is evidence that lower dietary diversity during pregnancy is directly associated with low birth weight, preterm birth, small for gestational age and infant mortality [45, 46].

The mean dietary diversity score of women in this study was lower compared to diversity scores of pregnant women from Bangladesh [22], Pakistan [47] and Kenya [21] but higher compared to scores of pregnant women from Ethiopia [48], and Malawi [49]. These differences may be due to differences in the study period i.e. season in which the data was collected, geographical location, and/or socio-cultural factors. Similarly, low dietary diversity scores have been reported by nationally representative studies conducted in Nepal (4.2 ± 1.4) [50] and in rural Nepal (4.6±1.2) [28]. Previous studies have reported the prevalence of anemia during pregnancy is more with low dietary diversity [51, 52]. This implies that about half of pregnant women may have micronutrient deficiencies. Also, the results display limited potential contribution of nutritious foods from food groups other than staples. Such inadequate dietary varieties in daily food consumption along with infrequent intake of seeds, vegetable, fruits, meat, and egg have been reported in the Terai [53] and rural Nepal [28], Bangladesh [22], and low and middle income countries [14] and are likely responsible for observed insufficient micronutrient intake and deficiencies in those settings [53, 54].

In contrast, a study done on secondary analysis of Nepal National Micronutrient Status Survey 2016 reported a 69.7% of non-pregnant women consuming meat and meat products [55]. The discordant findings between these national findings vs. the local findings from our study illustrate the need for further research to examine patterns within the country. During pregnancy, the regular inclusion of eggs in the diet in low and middle income countries can be crucial as a principle source of choline- a vital diet component essential for cell and brain development of the fetus and early infancy [56]. Policy makers should consider promoting egg production, consumption and year round consumption of vitamin A rich fruits and vegetables especially among the poor household and women of reproductive age. There are organizations like SUAHARA working in the district focusing on promoting dietary diversity among

mothers and children under 2 years of age but equal attention need to be given to newly pregnant women so all the target population could be given equal attention.

In this study, the diets of women who were more frequently involved in household decision making were positively associated with improved dietary diversity which is consistent from the finding from Northern Benin [57]. Similarly, an analysis of Ghana Demographic Health Survey revealed that the odds of achieving dietary diversity were higher among the women who had a say in deciding household purchases, compared to women who did not have a say [26]. Also, maternal dietary diversity is found to be higher among the women who have a control over their income in a study done in Nepal [58]. In the western region of Nepal most women do not have decision making opportunity. Women are found to be limited more to household activities that have low monetary value. Also, more women are dependent on the family and their husbands as they are more involved in care economy [59]. This suggests that program to improve women's nutrition should focus on increasing women's decision making power and their economic independency.

In this study, higher empowerment level that included women involved in cash earning activities (as per the definition of women empowerment in this study) was significantly associated with dietary diversity during pregnancy, a pattern that has also been among pregnant women in Kenya [21]. One possible explanation is that women involved in such activities have a regular income which increases their purchasing power and they are more likely to purchase varied food compared to pregnant women who do not earn. Similarly, our study found that education was positively associated with dietary diversity, a finding that has also been observed in Ghana [26], Bangladesh [22], Kenya [21] and Nepal [60]. This might be due to associations between education and socioeconomic status or knowledge.

This study revealed that women from wealthier households were more likely to consume diverse diet than women from poorer households which is supported by studies done in rural Bangladesh [23], Ghana [61], Kenya [21], Bangladesh, Vietnam, and Ethiopia [62] as well as a study conducted in the Terai region of Nepal, which found that socioeconomic status was positively associated with more frequent consumption of most food groups including in-season fruits and vegetables [27]. These findings support Bennett's law as with greater resources there is reallocation among food groups constituting diverse diet and consumption of improved and nutritious diets [63].

It is also seen that 30% of the participants prohibited the intake of certain food during pregnancy which is similar to the food restrictions (29%) reported in the Terai of Nepal, primarily influenced by cultural norms [64]. In this study, there weren't any significant differences in dietary intake among those practicing food taboos compared to those who did not. This is likely primarily because in this study most of the participants who did restrict food intake only did so in the first trimester of their pregnancy and at the time of interview, they were no longer practicing such food restrictions. Also, the major foods restricted like jackfruit was not available during the data collection period, reflecting seasonal variations in intake. It may also be attributed to the fact that women find alternative food items to replace restricted food from the same food group. Similarly, a study conducted in China also found no significant difference in nutrient intake among those practicing food taboos compared to those who did not during pregnancy [65].

Women with adequate knowledge of appropriate nutrition during pregnancy were more likely to consume diverse diet compared to women with no adequate knowledge which is consistent with the previous studies done in Bangladesh [66] and Ethiopia [67] and India [68]. These are encouraging findings in support of behavior change communication interventions during pregnancy that focus on adequate diets during this life stage.

Our study has some limitations. The cross-sectional nature of the data limits causal inference between dietary diversity and the correlates. Confinement of sample to a single season may limit the generalizability of findings to other seasons. The sampling frame was created with the help of FCHV and ANC registers of respective ward so there is the possibility of selection bias if women were left off the registers. The use of a dietary diversity score as a proxy for nutrient adequacy is also a limitation, even though it is widely used [69]. The findings can provide relevant information that can be used to inform nutrition sensitive interventions.

## Conclusion

The study showed that dietary diversity of the pregnant women in the hills of Nepal was low. It has also shown that only half of the pregnant women consumed diverse diet while the rest did not meet the minimum dietary diversity criteria. It can be concluded that greater access to resources through: earning capacity, wealth, education/ knowledge and, more empowered women, as measured by the study, was noted to be significantly associated with consumption of an adequately diverse diet during pregnancy. Hence public health policy and interventions aimed at promoting maternal dietary diversity should focus on promoting socioeconomic status, and providing environment for raising empowered women. In addition the responsible stake holders and health worker should strengthen the system and procedure for early counseling to impart knowledge on importance of dietary diversity for pregnancy and perinatal outcomes.

## Supporting information

**S1 Table. Food groups to construct Minimum Dietary Diversity for women of reproductive age (MDD-W).**
(DOCX)

**S2 Table. Variables included in assessing knowledge on nutrition during pregnancy.**
(DOCX)

**S3 Table. Variables included in construction of women empowerment index.**
(DOCX)

**S4 Table. VIF reports for the predictor variables.**
(DOCX)

**S1 File. Questionnaire.**
(DOCX)

**S1 Dataset.**
(ZIP)

## Acknowledgments

We would like to extend our special thanks to Assoc. Prof. Dr. Amod Kumar Poudyal for his continuous support during thesis process and data analysis and Central Department of Public Health, Institute of medicine for their consistent guidance and support. We would like to express our sincere thanks to Dr. Helen Harris Fry, Principal Investigator of the study on Development and Validation of a photographic food atlas for Nepal, for providing the photo atlas. We owe our utmost respect and thanks to the research participants for their active participation as well as to all the FCHVs and staff of the selected ward for their cooperation and support.

## Author Contributions

**Conceptualization:** Vintuna Shrestha, Swetha Manohar.

**Data curation:** Vintuna Shrestha.

**Formal analysis:** Vintuna Shrestha, Rajan Paudel, Dev Ram Sunuwar, Archana Amatya.

**Methodology:** Vintuna Shrestha.

**Resources:** Vintuna Shrestha.

**Software:** Vintuna Shrestha.

**Supervision:** Rajan Paudel, Dev Ram Sunuwar, Andrew L. Thorne Lyman, Swetha Manohar, Archana Amatya.

**Validation:** Vintuna Shrestha.

**Visualization:** Vintuna Shrestha.

**Writing – original draft:** Vintuna Shrestha.

**Writing – review & editing:** Vintuna Shrestha, Rajan Paudel, Dev Ram Sunuwar, Andrew L. Thorne Lyman, Swetha Manohar, Archana Amatya.

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
