## [Decision Letter · Decision Letter 0]

8 Dec 2020

PONE-D-20-29709

Factors associated with dietary diversity among pregnant women in the western hill region of Nepal: a community based cross-sectional study

PLOS ONE

Dear Dr. Shrestha,

Thank you for submitting your manuscript to PLOS ONE. After careful consideration, we feel that it has merit but does not fully meet PLOS ONE’s publication criteria as it currently stands. Therefore, we invite you to submit a revised version of the manuscript that addresses the points raised during the review process.

Overall, your manuscript is very interesting. However, it needs some improvements, particularly in introduction, discussions and conclusions sections. Please see reviewers' comments for details.

We look forward to receiving your revised manuscript.

Kind regards,

Abid Hussain

Academic Editor

PLOS ONE

Journal Requirements:

2. We note that Figure 1 in your submission contains map images which may be copyrighted.

We require you to either (a) present written permission from the copyright holder to publish these figure specifically under the CC BY 4.0 license, or (b) remove the figure from your submission:

b. If you are unable to obtain permission from the original copyright holder to publish these figure under the CC BY 4.0 license or if the copyright holder’s requirements are incompatible with the CC BY 4.0 license, please either i) remove the figure or ii) supply a replacement figure that complies with the CC BY 4.0 license. Please check copyright information on all replacement figures and update the figure caption with source information. If applicable, please specify in the figure caption text when a figure is similar but not identical to the original image and is therefore for illustrative purposes only.

3. Please ensure that you refer to Figure 1 in your text as, if accepted, production will need this reference to link the reader to the figure.

Additional Editor Comments:

Dear authors,

Thank you so much for your submission to PLOS ONE. There is need of further strengthening of introduction, discussions and conclusions sections. For details, please see reviewers' comments.

Reviewers' comments:

Reviewer's Responses to Questions

**Comments to the Author**

1. Is the manuscript technically sound, and do the data support the conclusions?

Reviewer #1: No

Reviewer #2: Yes

2. Has the statistical analysis been performed appropriately and rigorously? 

Reviewer #1: No

Reviewer #2: No

3. Have the authors made all data underlying the findings in their manuscript fully available?

Reviewer #1: No

Reviewer #2: Yes

4. Is the manuscript presented in an intelligible fashion and written in standard English?

Reviewer #1: Yes

Reviewer #2: Yes

5. Review Comments to the Author

Reviewer #1: This is an interesting paper examining dietary diversity among pregnant women. While the overall paper looks okay, I find nothing new in this paper. Most of the factors associated with dietary diversity are well established. One way, the paper could have benefited by contextualizing the measures which I did not see. This also lacked in the discussion. There is no discussion around policy implications of the study. Therefore, I don't fully understand what is the utility of this study. Perhaps, authors could have also examined the how MNCH outcomes among women are placed in relationship to the DD.

Aside from these broad remarks, following are some of the specific comments:

1. The authors says data are available within the manuscript. However, I did not find it. Perhaps, authors idea of data refers to tables. I would encourage authors to include the statement on where the data is available and how it can be accessed.

2. Reference to Figure 1 is missing in the text of the paper.

3. In the study context, provide more details on MNCH situation in Baglung and how it is different from rest of Nepal. It is important for international audience to understand the context better to interpret the findings.

4. Please provide more details on the enrollment process of the individuals into the study. You say that list of women were taken from FCHV? Was it not a violation of privacy of Individuals? How was the consent taken? What was the refusal rate?

5. Table 3: I feel something is not right about this table. Did you check for multi-collinearity. It seems several of the covariates may be correlated with each other. Also, it does not make sense to include a variable with very low cell frequencies. The authors could have combined such categories with another category of the variable.

6. For Table 3, present row percentages of dietary diversity. That would enable effective comparison. Also, check the p-value for food security in the multivariable analysis. It seems incorrect given the range of 95% CI.

7. In the discussion, the authors should interpret the findings more in the study context. I feel that is completely missing.

Good luck with the revision

Reviewer #2: The present study has explored one of the important issues among pregnant women in one of the urban areas in Nepal. Indeed, it’s an important issue to be discussed. The study is very interesting, I appreciate the efforts made by the authors for completing this project. My suggestions are mentioned below for further improvement.

1. As the study is explanatory in nature, therefore, I may suggest a strong theoretical framework. It will enrich the analysis and discussion.

2. The introduction section should have explanation of the factors that are associated with the dietary diversity from other parts of the world.

3. Study area selection, the sampling is well defined.

4. Why the variable is dichotomous, what is its advantage over the ratio variable and may be regressed as OLS method.

5. How the independent predictors are selected. Are these selected from the literature, expert opinion or by some other statistical techniques, please mention it.

6. Were these socio-economic factors checked for the multi-collinearity problem?

7. How the sample size assumption and linearity of independent variables and log odds are checked first.

8. In the conclusion section, please more focus on the policy implications. These implication should to the point and practically implemented. Moreover, what programs which are already going on in Nepal can help to solve the problems. These kind of problems may also be linked in discussion section with the results.

9. Language editing is needed.

6. PLOS authors have the option to publish the peer review history of their article (what does this mean?). If published, this will include your full peer review and any attached files.

Reviewer #1: No

Reviewer #2: **Yes: **Shahab E. Saqib

---

## [Author Response · Author response to Decision Letter 0]

21 Jan 2021

Academic editor comments and concern 

We note that Figure 1 in your submission contains map images which may be copyrighted.

AUTHORS: Thank you very much for your concern. We assure you that the provided Fig. 1 contains a study area Map is copyright free. We created this map using Arc GIS software version 10.8 and base files of the administrative provinces and districts of Nepal were obtained from the freely available copyright free resources Government of Nepal, Ministry of Land Management, [1] and Natural Earth.[2]. The map was displayed the location of the study sites in Baglung Municipality, Baglung district, Nepal. 

Please ensure that you refer to Figure 1 in your text as, if accepted, production will need this reference to link the reader to the figure.

AUTHORS: Thank you for the comments. We have mentioned the Fig 1 legend as “Fig.1 Map showing the location of study sites" in the revised manuscript in line number: 96 

Your ethics statement should only appear in the Methods section of your manuscript. If your ethics statement is written in any section besides the Methods, please delete it from any other section.

AUTHORS: Thank you for kind suggestion. We have deleted the ethical statement in declaration section in the revised manuscript.

Reviewer 1

This is an interesting paper examining dietary diversity among pregnant women. While the overall paper looks okay, I find nothing new in this paper. Most of the factors associated with dietary diversity are well established. One way, the paper could have benefited by contextualizing the measures which I did not see. This also lacked in the discussion. There is no discussion around policy implications of the study. Therefore, I don't fully understand what the utility of this study is. Perhaps, authors could have also examined the how MNCH outcomes among women are placed in relationship to the DD.

AUTHORS: Thank you very much for your comments. We feel it’s important to place the study into context here: indeed, factors associated with dietary diversity among women of reproductive age are fairly well established in the international context. In Nepal, however, only limited studies have been published; the country is very diverse in terms of both its ecology and cultural practices. To date most of the published studies were conducted in the Terai region of the country where two active research sites exist. Comparably little information is available on dietary diversity among pregnant women in the hill region of Nepal. Thus, we feel this study provides an important source of information on dietary practices in this region which may differ from findings from other regions. It is also important to note that we have focused on pregnant women specifically, which is important because cultural practices in pregnancy in Nepal influence the foods that women eat (food taboos, prescriptions and proscriptions are present). We have added the manuscript to better emphasize the context and importance. 

The authors says data are available within the manuscript. However, I did not find it. Perhaps, authors idea of data refers to tables. I would encourage authors to include the statement on where the data is available and how it can be accessed. 

AUTHORS: We uploaded the data to accompany the manuscript during the manuscript submission process. The filename is Dataset.dta which is zip/ compressed file. We have again uploaded the file “Dataset.dta” under the item section Supporting Information ZIP files. 

Reference to Figure 1 is missing in the text of the paper.

AUTHORS: Thank you for pointing this out. We have added this. 

In the study context, provide more details on MNCH situation in Baglung and how it is different from rest of Nepal. It is important for international audience to understand the context better to interpret the findings.

AUTHORS: This is important feedback. We have added the information under the subheading study design and setting in lines: 91-95, page number: 5. We have included statistics on low birth weight, ANC visits and iron supplementation. Although we wanted to add more information like maternal underweight, could not collect it even after rigorous data searching in internet and study of annual report of Baglung.

Please provide more details on the enrollment process of the individuals into the study. You say that list of women were taken from FCHV? Was it not a violation of privacy of Individuals? How was the consent taken? What was the refusal rate?

AUTHORS: Ethical approval was obtained for the study from the Institute of Medicine at Tribhuvan University and written consent was obtained from all participants as described under the ethical consideration subheading, lines: 190-195, page number: 9. An approval letter introducing the researcher and stating the study’s objectives was shared with Baglung Municipality’s Health Section Chief and its Mayor as well as the responsible supervisor of the health posts in each of the selected wards. Formal permission was taken from these concerned authorities. The refusal rate was zero, which is not uncommon in Nepal where research participation is typically high [3]. 

Table 3: I feel something is not right about this table. Did you check for multi-collinearity. It seems several of the covariates may be correlated with each other. Also, it does not make sense to include a variable with very low cell frequencies. The authors could have combined such categories with another category of the variable.

AUTHORS: We have checked multi-collinearity and report of the VIF was included in supplementary file as “S4 Table” in the revised manuscript. Initially, a total of 19 predictor variables were included in our study. We agree with you with regards to some variables not providing a great deal of additional discriminatory information such as religion, first ANC visit, and food security. As such, we have removed these variables from the Table 3 and now a total of 16 variables are included in the revised manuscript. In addition, we have merged the women empowerment variable where we noticed that one cell have low cell frequency. Women empowerment was classified as ‘low’, ‘moderate’, and ‘high’. Now we have merged low/ moderate and present the data in the revised manuscript.

For Table 3, present row percentages of dietary diversity. That would enable effective comparison. Also, check the p-value for food security in the multivariable analysis. It seems incorrect given the range of 95% CI. 

AUTHORS: Thank you for these suggestions. We have revised the Table 3 and presented row percentage of dietary diversity and added the related statistics in the revised manuscript.

In the discussion, the authors should interpret the findings more in the study context. I feel that is completely missing.

AUTHORS: We have added the more contextual information in the discussion section within the revised manuscript, line: 266-267, page number: 16; line: 281-286, page number: 17; line: 294-298, page number: 17-18 and line: 306, page number: 18.

Reviewer #2: The present study has explored one of the important issues among pregnant women in one of the urban areas in Nepal. Indeed, it’s an important issue to be discussed. The study is very interesting, I appreciate the efforts made by the authors for completing this project. 

AUTHORS: We highly appreciate your thorough review. 

As the study is explanatory in nature, therefore, I may suggest a strong theoretical framework. It will enrich the analysis and discussion.

AUTHORS: Thank you for the suggestion. We have added this to the paper in Fig 2. Legend as “Fig.2 Conceptual framework on factors associated with dietary diversity” in line number: 172. 

The introduction section should have explanation of the factors that are associated with the dietary diversity from other parts of the world.

AUTHORS: We have added to line number: 72-74, page number: 4 to highlight information on factors associated with dietary diversity from other parts of the world in the introduction section.

Study area selection, the sampling is well defined.

AUTHORS: We thank you for the comment.

Why the variable is dichotomous, what is its advantage over the ratio variable and may be regressed as OLS method.

AUTHORS: The dependent variable dietary diversity is dichotomous as it was used from Minimum dietary diversity for women of reproductive age (MDD-W) indicator developed by Food and Agriculture Organization of the United Nation (FAO). MDD-W is a population level indicator that can be used as a proxy indicator for higher micronutrient adequacy, one important dimension of diet quality. MDD-W is a dichotomous indicator that classifies women into those with low dietary diversity (who would be highly unlikely to meet their micronutrient requirements) and those with minimum dietary diversity (who would be more likely to meet their micronutrient requirements) so it signifies whether micronutrients are met or not. Thus being a dichotomous variable it’s easier to interpret, communicate and advocate even to individual with no nutrition knowledge rather than continuous variable. 

How the independent predictors are selected. Are these selected from the literature, expert opinion or by some other statistical techniques, please mention it.

AUTHORS: Independent variables were selected after intensive literature review

Were these socio-economic factors checked for the multi-collinearity problem?

AUTHORS: Yes, the variables were checked for the multi-collinearity problem and reports are presented in supplementary file as S4 Table.

How the sample size assumption and linearity of independent variables and log odds are checked first.

AUTHORS: To check for linearity of independent variables multicollinearity test was done. None of the variables had Variance Inflation Factors (VIF) more than 10. There was no problem of collinearity among independent variables as the highest VIF was 1.9.

In the conclusion section, please more focus on the policy implications. These implication should to the point and practically implemented. Moreover, what programs which are already going on in Nepal can help to solve the problems. These kind of problems may also be linked in discussion section with the results.

AUTHORS: Thank you for your comment. We have added the line: 347-349, page number: 20 to focus on practical implications of the finding of the study.

Language editing is needed.

AUTHORS: We have gone through the manuscript again to try our best and edit the language where needed.

References 

1. Goverment of Nepal, Survey Department. Ministry of Land Management, Coperatives and Poverty Alleviation. 2020. 

2. North American Cartographic Information Society (NACIS) Collaborators. Natural Earth. 

3. Karmacharya C, Saleh A, Shrestha B, Bhandari S, Manohar S, Thorne-Lyman A, et al. PoSHAN Community Studies: Panel 2 Survey Report. 2016.

---

## [Editor Report · Decision Letter 1]

2 Feb 2021

Factors associated with dietary diversity among pregnant women in the western hill region of Nepal: a community based cross-sectional study

PONE-D-20-29709R1

Dear Dr. Shrestha,

We’re pleased to inform you that your manuscript has been judged scientifically suitable for publication and will be formally accepted for publication once it meets all outstanding technical requirements.

Kind regards,

Abid Hussain

Academic Editor

PLOS ONE

Additional Editor Comments (optional):

Dear Authors,

Thank you for adequately addressing the comments from the reviewers.
---

## [Editor Report · Acceptance letter]

30 Mar 2021

PONE-D-20-29709R1 

Factors associated with dietary diversity among pregnant women in the western hill region of Nepal: a community based cross-sectional study 

Dear Dr. Shrestha:

I'm pleased to inform you that your manuscript has been deemed suitable for publication in PLOS ONE. Congratulations! Your manuscript is now with our production department. 

Kind regards, 

on behalf of

Dr. Abid Hussain 

Academic Editor

PLOS ONE